Selection of reference genes for quantitative real-time PCR analysis in cucumber (Cucumis sativus L.), pumpkin (Cucurbita moschata Duch.) and cucumber–pumpkin grafted plants

Miao Li 1 2
Qin Xing 3
Gao Lihong 2
Li Qing 1
Li Shuzhen 1
He Chaoxing 1
Li Yansu liyansu@caas.cn 1
Yu Xianchang yuxianchang@caas.cn 1
1 Institute of Vegetables and Flowers, Chinese Academy of Agricultural Sciences , Beijing , China
2 Beijing Key Laboratory of Growth and Developmental Regulation for Protected Vegetable Crops, College of Horticulture, China Agricultural University , Beijing , China
3 Agricultural Genomics Institute at Shenzhen, Chinese Academy of Agricultural Sciences , Shenzhen , China
Cuypers Ann
Electronic publication date: 2019 Apr 17
Publication date: 2019
Volume: 7
Electronic Location ID: e6536
Received 2018 Oct 29; Accepted 2019 Jan 29
Copyright: ©2019 Miao et al.
Copyright year: 2019
Copyright holder: Miao et al.
License: This is an open access article distributed under the terms of the Creative Commons Attribution License, which permits unrestricted use, distribution, reproduction and adaptation in any medium and for any purpose provided that it is properly attributed. For attribution, the original author(s), title, publication source (PeerJ) and either DOI or URL of the article must be cited.
License URL: https://creativecommons.org/licenses/by/4.0/

Keywords: Cucumber, Pumpkin, Gene expression, Grafted cucumber, Quantitative real-time PCR, Reference genes

Funding: National Natural Science Foundation of China No. 31772363 Science and Technology Innovation Program of the Chinese Academy of Agricultural Sciences (CAAS-ASTIP-IVFCAAS) Key Laboratory of Horticultural Crop Biology and Germplasm Innovation, Ministry of Agriculture, China Special Project of Basic Research Expenses of Public Welfare Research Institute of Vegetables and Flowers of the Chinese Academy of Agricultural Sciences IVF-BRF2018013 This research was supported by the National Natural Science Foundation of China (No. 31772363), the Science and Technology Innovation Program of the Chinese Academy of Agricultural Sciences (CAAS-ASTIP-IVFCAAS), the Key Laboratory of Horticultural Crop Biology and Germplasm Innovation, Ministry of Agriculture, China, and the Special Project of Basic Research Expenses of Public Welfare Research Institute of Vegetables and Flowers of the Chinese Academy of Agricultural Sciences (IVF-BRF2018013). The funders had no role in study design, data collection and analysis, decision to publish, or preparation of the manuscript.

==============================
Background

Quantitative real-time PCR (qRT-PCR) is a commonly used high-throughput technique to measure mRNA transcript levels. The accuracy of this evaluation of gene expression depends on the use of optimal reference genes. Cucumber–pumpkin grafted plants, made by grafting a cucumber scion onto pumpkin rootstock, are superior to either parent plant, as grafting conveys many advantages. However, although many reliable reference genes have been identified in both cucumber and pumpkin, none have been obtained for cucumber–pumpkin grafted plants.

Methods

In this work, 12 candidate reference genes, including eight traditional genes and four novel genes identified from our transcriptome data, were selected to assess their expression stability. Their expression levels in 25 samples, including three cucumber and three pumpkin samples from different organs, and 19 cucumber–pumpkin grafted samples from different organs, conditions, and varieties, were analyzed by qRT-PCR, and the stability of their expression was assessed by the comparative ΔCt method, geNorm, NormFinder, BestKeeper, and RefFinder.

Results

The results showed that the most suitable reference gene varied dependent on the organs, conditions, and varieties. CACS and 40SRPS8 were the most stable reference genes for all samples in our research. TIP41 and CACS showed the most stable expression in different cucumber organs, TIP41 and PP2A were the optimal reference genes in pumpkin organs, and CACS and 40SRPS8 were the most stable genes in all grafted cucumber samples. However, the optimal reference gene varied under different conditions. CACS and 40SRPS8 were the best combination of genes in different organs of cucumber–pumpkin grafted plants, TUA and RPL36Aa were the most stable in the graft union under cold stress, LEA26 and ARF showed the most stable expression in the graft union during the healing process, and TIP41 and PP2A were the most stable across different varieties of cucumber–pumpkin grafted plants. The use of LEA26, ARF and LEA26+ARF as reference genes were further verified by analyzing the expression levels of csaCYCD3;1, csaRUL, cmoRUL, and cmoPIN in the graft union at different time points after grafting.

Discussion

This work is the first report of appropriate reference genes in grafted cucumber plants and provides useful information for the study of gene expression and molecular mechanisms in cucumber–pumpkin grafted plants.

Introduction

Cucumber (Cucumis sativus L.) is one of the most widely cultivated vegetable crops in the world. Grafted cucumber plants are popular due to their greater resistance to soil-borne diseases, increased tolerance to abiotic stress, improved mineral nutrition uptake and use, and increased fruit yield and quality (Huang et al., 2014). A cucumber scion is typically grafted onto pumpkin (Cucurbita moschata Duch.) rootstock (Lee et al., 2010; Huang et al., 2014). Grafting conveys advantages over the properties of each individual parent plant, but the resulting plant is also more complex than the parents. The graft union is the successful combination of the scion and rootstock, allowing the establishment of complex communication between rootstock and scion. There have been many physiological and biochemical studies of cucumber–pumpkin grafted plants (Ahn et al., 1999; Yang et al., 2006; Haroldsen et al., 2012; Li et al., 2014), however, there have been few studies analyzing gene function, transcription, or expression in cucumber–pumpkin grafted plants, as few pumpkin genes were identified. Now, the entire cucumber (http://cucurbitgenomics.org/organism/2) (Huang et al., 2009) and pumpkin (http://cucurbitgenomics.org/organism/9) (Sun et al., 2017) genomes have been published, enabling further studies on the molecular biology of these species.

Gene expression analysis is fundamental to elucidate the molecular mechanisms underlying various biological processes (Bustin et al., 2005), and qRT-PCR is the most common technique used to study gene expression because of its high sensitivity, accuracy, specificity, cost-effectiveness, and reproducibility (Bustin, 2002; Nolan, Hands & Bustin, 2006; Derveaux, Vandesompele & Hellemans, 2010). However, non-specific variations can cause errors resulting in unreliability of the qRT-PCR data, such as variability in RNA quality, cDNA synthesis and concentration, PCR procedures, and efficiency of amplification (Delporte et al., 2015). To avoid these problems in analyzing results, stable reference genes should be used to normalize the gene expression data. Appropriate reference genes should be systematically evaluated across various environments (varieties, tissues, experimental treatments, and developmental stages) before being used as controls in qRT-PCR analysis (Bustin et al., 2009; Guénin et al., 2009; Sgamma et al., 2016). However, there are no previous reports of systematic studies performed on grafted cucumber plants to determine reliable reference genes.

Common reference genes like ACT (actin), TUA (tubulin), CYP (cyclophilin), UBI-1 (ubiquitin), and EF- α (elongation factor) are considered to be stably expressed in most plants (Duan et al., 2017; Obrero et al., 2011; Tashiro, Philips & Winefield, 2016; Niu et al., 2017) and have been used for gene expression studies in cucumber (Wan et al., 2010; Migocka & Papierniak, 2011; Warzybok & Migocka, 2013). The genes UFP (ubiquitin), EF-1A (elongation factor), PRL36aA (60S ribosomal protein L36a/L44), PP2A (protein phosphatase) and CACS (clathrin adaptor complexes medium submit family protein) have been used for reliable normalization in different experimental sets in zucchini (Cucurbita pepo) (Obrero et al., 2011), and these reference genes have been successfully applied to both cucumber and pumpkin in specific environments, including powdery mildew, salinity, cold, dehydration, H2O2, and abscisic acid (ABA) treatments (Berg et al., 2015; Cao et al., 2017; Reda et al., 2018). Unfortunately, no single reference gene has been confirmed that exhibits uniform and stable expression under different experimental conditions. For example, ACT is a frequently used reference genes in many plants, but showed variant expression during short-term treatment of cucumber with salt, osmotic, or oxidative stress (Migocka & Papierniak, 2011). Overall, it is necessary to identify one or more reference genes that show stable expression under different experimental conditions prior to carrying out gene expression studies (Duan et al., 2017).

In this study, traditional reference genes from published research and new genes based upon their coefficients of variation (CVs) and expression intensity in our RNA-seq data from cucumber–pumpkin grafted plants at different stages were selected for further analysis. Twelve genes were investigated in this study, eight commonly used reference genes, ACT, CYP, CACS, TUA, TIP41 (tonoplast intrinsic protein), F-Box (F-box protein), RPL36Aa, and PP2A, and four new genes identified by RNA-seq analysis, UBC (Ubiquitin conjugating enzyme), ARF (ADP-ribosylation factor-like protein), LEA26 (Late-embryogenesis abundant protein 26), and 40SRPS8 (40S ribosomal protein S8). These twelve genes were evaluated to validate their use as stable reference genes for qRT-PCR in different organs, at different stages, in different varieties, and under stress conditions in cucumber, pumpkin, and cucumber–pumpkin grafted plants. To determine the appropriate reference genes, four statistical tools were used to evaluate the accuracy of these candidate genes: the ΔCt method (Silver et al., 2006), geNorm (Vandesompele et al., 2002), NormFinder (Andersen, Jensen & Orntoft, 2004), and BestKeeper (Pfaffl et al., 2004). Comprehensive stability rankings were generated by RefFinder (Xie et al., 2012). Additionally, the expression of CYCD3;1, RUL and PIN, genes that are thought to be related to graft union healing in Arabidopsis (Melnyk et al., 2018), were investigated as a case study to evaluate the effectiveness of the reference genes identified in this study. The results obtained in this study will be useful in future gene expression analyses in cucumber, pumpkin, and their grafted plants.

Materials & Methods

Plant materials and treatments

Cucumber (Cucumis sativus L.) and pumpkin (Cucurbita moschata Duch.) were planted in an artificial chamber at the farm of the Institute of Vegetables and Flowers, Chinese Academy of Agricultural Sciences, Beijing, China at a temperature of 28 °C/20 °C (day/night) with a photoperiod cycle of 12/12 h and 60%–70% relative humidity. Cucumber variety ‘Zhongnong No. 26′was used as the scion and pumpkin variety ‘Jinxinzhen No. 5′was used as the rootstock. Seeds of the scion and rootstock were sown in 50-cell (55 cm3/cell) and 32-cell (32 cm3/cell) polystyrene trays, respectively, containing commercial organic substrates (Vpeatmoss:Vvermiculite:Vperlite = 1:1:1). The environmental conditions for germination were 25–28 °C and 85%–90% relative humidity. The pumpkin seeds were sown three days before the cucumber seeds. When cotyledons of the scion were fully open and the first true leaf of the rootstock started to develop (9–10 d after sowing), the plants were grafted using the hole insertion grafting method as previously described (Miao et al., 2018) (Fig. S1). Autografts were carried out for both cucumber and pumpkin, as well as cucumber–pumpkin heterografts. The grafted seedlings were maintained at a temperature of 30 °C/22 °C (day/night), a constant humidity of 95%–100% and a dim light of 50 PPFD (photosynthetic photon flux density) for the first 5 days, then the light density was slowly increased from 50 to 500 PPFD and the humidity was decreased from 95% to 60%, while the other environmental conditions were unchanged. For the autograft cucumber and pumpkin plants, samples of the leaves, stems and roots were harvested when the seedlings had two true leaves. For cucumber grafted onto pumpkin, samples of the leaves, the stem of the scion, the graft union (Fig. 1), the stem of the rootstock, and the roots were harvested. For the cold stress experiment, when the grafted cucumber had two leaves, seedlings were exposed to temperatures of 12 °C in a chamber, and samples of the graft union were harvested at 0, 5, 12 and 24 h of stress treatment. To investigate the graft union healing process, samples of the graft union were harvested 0, 3, 6, 9 and 15 d after grafting. For experiments with varieties, cucumber varieties ‘Xintaimici’ and ‘Zhongnong No. 26′were used as scions and pumpkin varieties ‘Zhongguonangua No. 26′, ‘Jinxinzhen No. 5′and ‘Huofenghuang’ were used as rootstocks. The graft combinations were ‘Xintaimici–Zhongguonangua No. 26′, ‘Xintaimici–Jinxinzhen No. 5′, ‘Xintaimici–Huofenghuang’, ‘Zhongnong No. 26–Jinxinzhen No. 5′, and ‘Zhongnong No.26–Huofenghuang’. Graft unions were harvested when grafted plants had two true leaves. For each treatment, three independent biological replicates were achieved. All samples were immediately frozen in liquid nitrogen and stored at −80 °C.

Figure 1 Graft union of cucumber-pumpkin grafted plants.

Red box indicates graft union of cucumber-pumpkin grafted plant, and the upper is scion-cucumber, the lower part is rootstock-pumpkin. A cucumber cultivar (Zhongnong No.26)was used as the scion, a pumpkin cultivar (Jinxinzhen No.5) was used as the rootstock. Graft union of cucumber-pumpkin grafted plants 20 d after grafting. Red box indicates graft union of cucumber-pumpkin grafted plant, and the upper is scion-cucumber, the lower part is rootstock-pumpkin.

RNA isolation and cDNA synthesis

The RNAprep Pure Plant Plus Kit (Tiangen, Beijing, China) was used for total RNA extraction. Genomic DNA was eliminated from the total RNA using RNase-free DNase I. The RNA integrity was confirmed by 1.0% agarose gel electrophoresis. RNA concentrations were determined by NanoDrop™ 2000 spectrophotometer (Thermo Scientific, Waltham, MA, USA), samples with an A260/A280 ratio of 1.8–2.2 and an A260/A230 ratio >2.0 were used for further analyses. First-strand cDNA synthesis was performed using a FastQuant cDNA Synthesis kit (Tiangen, Beijing, China) according to the manufacturer’s instructions.

Candidate reference gene selection and primer design

Eight commonly used reference genes (ACT, CYP, CACS, TUA, TIP41, F-Box, PRL36Aa and PP2A) from published studies on cucumber, pumpkin, chicory, buckwheat, lettuce and mangrove tree were selected (Wan et al., 2010; Obrero et al., 2011; Delporte et al., 2015; Demidenko, Logacheva & Penin, 2011; Borowski et al., 2014; Saddhe, Malvankar & Kumar, 2018). For new candidate reference genes, we analyzed our transcriptomic data from the graft union. Graft union of cucumber-pumpkin were respectively harvested at 0, 3, 6, 9 days after grafting, three biological replicates were performed for each time point. In total 18 transcriptome libraries, 132.7G raw reads were obtained, at the least 91.4% of the reads were mapped to the reference sequence, and assembled into 32,852 and 47,906 transcripts of cucumber and pumpkin, respectively. Assemblies resulted in 20,782 unigene for cucumber with the average length of 4.1kb obtained, while 27,187 unigene with average length of 4.4kb were generated for pumpkin (Table S1). The genes with the most consistent expression levels were defined as candidate reference genes (De Jonge et al., 2007). We calculated the mean expression value, standard deviation, and coefficients of variation (CVs) based on the raw RNA-seq data, and CVs = standard deviation of RPKM/average of RPKM. Based on the requirements CV ≤ 0.2 and 300 ≤RPKM ≤ 500 (Duan et al., 2017), we selected new reference genes by removing overabundant genes with low expression levels. With requirements of evalue e-5, we used BLAST (Basic Local Alignment Search Tool) to determine the proteins encoded by cucumber and pumpkin genes, respectively (https://blast.ncbi.nlm.nih.gov/Blast.cgi), and then filtered the BLAST results based on an identity ≥ 90 and an overlap ratio >0.5 (between query and target). This resulted in ten and seven genes of cucumber and pumpkin, respectively, which were deemed suitable as reference genes. A comparison of the relationship between cucumber and pumpkin by homology analysis is shown in Table S2. Finally, UBC, ARF, LEA26 and 40SRPS8 were selected as candidate reference genes based on preliminary experiments of single PCR product in agarose gel electrophoresis (Fig. S2). Based on the conserved sequence of these genes between cucumber and pumpkin, primers were designed using Primer Premier 5.0 software with the following parameters: a melting temperature (Tm) of 50–60 °C, a primer length of 17–25 bp, and a product size of 70–260 bp (http://www.premierbiosoft.com/) (Table 1). Amplification of a single PCR product in 1% agarose gel electrophoresis and a single peak of the melting curve in qRT-PCR were used to ensure the specificity of the primers for the candidate reference genes.

Table 1 Description of the candidate references, primer sequences and RT-PCR amplification efficiencies in cucumber, pumpkin, and grafted cucumber/pumpkin.

Gene	Accession number (NCBI)	Annotation	Gene ID in cucumber	Forward primer (5′–3′)	Reverser primer (5′–3′)	Amplification length	Tm(°C)	RT-qPCR efficiency	
								Cucumber	Pumpkin	Cucumber/ pumpkin	
ACT	AB010922	Actin (ACT)	Csa6G484600	TCTCCGTTTGGACCTTGC	ATTTCCCGTTCGGCAGT	99	83.2	0.88	1.05	0.86	
CYP	AY942800	Cyclophilin	Csa7G009740	TTTCATGTGCCAGGGAGG	AGCCAATCGGTCTTAGCG	189	88.1	0.99	1.05	1.05	
CACS	GW881874	Clathrin adaptor complex subunit (CACS)	Csa3G902930	TGGGAAGATTCTTATGAAGTGC	CTCGTCAAATTTACACATTGGT	171	84.2	1.02	0.95	1.00	
TUA	AJ715498	Alpha-tubulin (TUA)	Csa4G000580	TCAGCGGCAAGGAAGATG	GCGGATTCTGTCCAAGCA	92	83.7	1.03	0.87	1.00	
TIP41	GW881871	TIP41-like family protein	Csa7G071610	TGGGAGGATTGCGAGGAGA	AAGTGATATGCCATTGTCAGC	117	81.6	0.97	1.08	1.13	
F-BOX	GW881870	F-box/kelch-repeat protein	Csa5G642160	TGGTTCATCTGGTGGTCTTG	TTAGCTGCCTCTGCTGATTG	131	84.3	1.08	0.93	0.90	
PRL36Aa	HM594174	60S ribosomal protein L36a/L44	Csa3G653380	AAGATAGTCTTGCTGCACAGGG	AACACGGGCTTGGTTTGA	79	83.3	0.97	0.95	0.99	
PP2A	HM594171	protein phosphatase 2A regulatory subunit A	Csa5G608520	GAAGCTGTAGGACCTGAACCA	AGCCGCTGCAATACGAAC	96	84.6	1.07	1.13	0.91	
UBC	–	–	Csa3G358610	GTCACCATTCATTTTCCTCCG	GGGCTCCACTGCTCTTTCA	131	83.9	1.04	1.07	1.12	
ARF	–	–	Csa5G524710	CTGCTGGAAAGACCACGAT	GACCACCAACATCCCATACA	132	83.5	1.02	1.12	1.03	
LEA26	–	–	Csa2G151040	CGTTGACTTACCCATCACCTTC	GCGTGTAGTACCACCCTCTTTA	163	85.5	1.00	1.06	0.98	
40SRPS8	–	–	Csa6G382970	ACTCGACACTGGAAACTACTCG	CCTGAACAACGGCACTCTT	134	85.1	0.87	1.03	1.01	

qRT-PCR assay

qRT-PCR was performed on an Agilent Stratagene Mx3000P Real-Time PCR machine (Agilent Stratagene, Santa Clara, CA, USA) using SYBR® Premix Ex Taq™ (TliRNaseH Plus) (TaKaRa, Dalian, China). Each 20 µl reaction mixture contained 2 µl of cDNA template, 0.4 µl of each primer, 0.4 µl of ROX dye, 10 µl of 2 × SYBR Premix Ex Taqand 6.8 µl of ddH2O. The qRT-PCR reaction conditions were as follows: 94 °C for 30 s, 40 cycles of 94 °C for 5 s, then 60 °C for 34 s. A melting curve was determined by increasing the amplification temperature from 60–95 °C, with a temperature increment of 0.5 °C every 5 s. All samples were performed with three technical replicates, and samples without template were used as a control. The amplification efficiencies for each primer and the regression coefficients (R2) were evaluated using five-fold dilutions of pooled cDNA (1/5, 1/25, 1/125, 1/625, 1/3125) that were diluted using EASY dilution solution (Takara, Kusatsu, Japan).

Gene expression stability analysis

To evaluate the expression levels of each reference gene, we drew boxplots of the Ct values for the 12 candidate reference genes. Four statistical tools, the ΔCt method (Silver et al., 2006), geNorm (Vandesompele et al., 2002), NormFinder (Andersen, Jensen & Orntoft, 2004), and BestKeeper (Pfaffl et al., 2004), were used to evaluate the stability of the 12 candidate reference genes at various treatment durations. The raw Ct values of the reference genes were transformed into the correct input files according to the requirements of the software. Finally, a comprehensive ranking of the reference genes was generated using RefFinder (Duan et al., 2017).

Validation of reference gene stability

To confirm the reliability of the selected reference genes, the relative expression levels of three genes involved in xylem development were measured during graft union healing in grafted cucumbers (Table S3). Samples of the graft union of cucumber–pumpkin grafted plants were harvested at 0, 1, 3, 6, 9 and 15d after grafting. The most stable reference genes (LEA26, ARF and LEA26 +ARF), and the least stable reference gene (PP2A) ranked by RefFinder were used for normalization. Comparative gene expression levels of csaCYCD3;1 (Csa2G356610), csaRUL (Csa3G895630), cmoRUL (CmoCh15G013320) and cmoPIN (CmoCh15G009810) were calculated using the 2−ΔΔCt method (Livak & Schmittgen, 2001). Three technical replicates were performed for each biological sample.

Results

Evaluation of primer specificity and amplification efficiency

To validate the primer specificity of the 12 candidate reference genes in our study, the specificity of the PCR reactions was subjected to 1% agarose gel electrophoresis (Fig. S2). The product lengths were consistent with the expected lengths, and a single sharp peak was observed in the melting curves for cucumber, pumpkin, and grafted cucumber (Figs. S3, S4A). Additionally, the amplification efficiency (E) ranged from 0.86 to 1.13, with the correlation coefficients (R2) of the standard curve varying from 0.986 to 0.999 (Table 1, Figs. S4B and S5).

Figure 2 Ct values of 12 candidate reference genes from the qRT-PCR analysis in all samples.

Boxplots show the 25th and 75th percentiles, means, and outliers. For each reference gene, the line inside the box is the means. The top and bottom line of the box are 75th and 25th percentiles. The circles above or below the box are outliers.

Expression levels and variations in candidate reference genes

The transcript abundances of the 12 candidate reference genes were assessed based on the Ct values from the qRT-PCR for different kinds of samples. As shown in Fig. 2, the Ct values for the 12 candidate reference genes ranged from 16.98 to 31.71, and the mean Ct values were 19.04, 18.35, 23.26, 20.80, 20.66, 26.70, 20.79, 24.79, 20.26, 21.78, 20.8 and 21.09 for ACT, CYP, CACS, TUA, TIP41, F-Box, PRL36Aa, PP2A, UBC, ARF, LEA26 and 40SRPS8, respectively.

Table 2 Overall ranking of the candidate reference genes in eight groups by ΔCt method, BestKeeper, NormFinder, geNorm, and RefFinder.

Method	1	2	3	4	5	6	7	8	9	10	11	12	
Ranking Order of candidate reference genes in different organs of cucumber plants (Better–Good–Average)	
Delta CT	TIP41	CACS	40SRPS8	TUA	PP2A	CYP	UBC	RPL36Aa	ACT	ARF	LEA26	F-Box	
BestKeeper	CACS	TIP41	40SRPS8	PP2A	TUA	RPL36Aa	CYP	ARF	F-Box	ACT	LEA26	UBC	
Normfinder	TIP41	CACS	40SRPS8	PP2A	TUA	CYP	ARF	RPL36Aa	UBC	ACT	LEA26	F-Box	
geNorm	TIP41 — 40SRPS8	CACS	TUA	PP2A	CYP	UBC	ACT	ARF	LEA26	RPL36Aa	F-Box	
Recommended comprehensive ranking	TIP41	CACS	40SRPS8	PP2A	TUA	CYP	RPL36Aa	ARF	UBC	ACT	LEA26	F-Box	
Ranking Order of candidate reference genes in different organs of pumpkin plants (Better–Good–Average)	
Delta CT	TIP41	PP2A	UBC	F-Box	ARF	CACS	CYP	ACT	40SRPS8	RPL36Aa	LEA26	TUA	
BestKeeper 	TIP41	UBC	PP2A	F-Box	ARF	CYP	ACT	CACS	LEA26	40SRPS8	RPL36Aa	TUA	
Normfinder	TIP41	PP2A	UBC	F-Box	ARF	CYP	ACT	CACS	40SRPS8	RPL36Aa	LEA26	TUA	
geNorm	CACS — 40SRPS8	RPL36Aa	PP2A	TIP41	UBC	ACT	CYP	F-Box	ARF	LEA26	TUA	
Recommended comprehensive ranking	TIP41	PP2A	UBC	CACS	F-Box	40SRPS8	ARF	CYP	ACT	RPL36Aa	LEA26	TUA	
Ranking Order of candidate reference genes in different organs of cucumber/pumpkin grafted plants (Better–Good–Average)	
Delta CT	CACS	40SRPS8	ARF	CYP	TUA	RPL36Aa	UBC	TIP41	LEA26	F-Box	ACT	PP2A	
BestKeeper 	CYP	RPL36Aa	40SRPS8	ARF	CACS	LEA26	UBC	TUA	ACT	TIP41	F-Box	PP2A	
Normfinder	40SRPS8	CACS	ARF	TUA	CYP	RPL36Aa	TIP41	F-Box	UBC	LEA26	ACT	PP2A	
geNorm	CACS — ARF	40SRPS8	CYP	RPL36Aa	TUA	UBC	LEA26	ACT	TIP41	F-Box	PP2A	
Recommended comprehensive ranking	CACS	40SRPS8	ARF	CYP	RPL36Aa	TUA	UBC	LEA26	TIP41	F-Box	ACT	PP2A	
Ranking Order of candidate reference genes in graft union of cucumber/pumpkin plants under low temperature (Better–Good–Average)	
Delta CT	TUA	CACS	RPL36Aa	F-Box	40SRPS8	CYP	ARF	ACT	UBC	LEA26	PP2A	TIP41	
BestKeeper 	TUA	RPL36Aa	CACS	CYP	40SRPS8	F-Box	ACT	ARF	UBC	PP2A	TIP41	LEA26	
Normfinder	TUA	CACS	RPL36Aa	F-Box	40SRPS8	ARF	ACT	CYP	UBC	LEA26	PP2A	TIP41	
geNorm	CYP — UBC	40SRPS8	RPL36Aa	TUA	ACT	CACS	F-Box	ARF	LEA26	PP2A	TIP41	
Recommended comprehensive ranking	TUA	RPL36Aa	CACS	CYP	40SRPS8	UBC	F-Box	ACT	ARF	LEA26	PP2A	TIP41	
Ranking Order of candidate reference genes in graft union during healing process (Better–Good–Average)	
Delta CT	LEA26	F-Box	TIP41	40SRPS8	RPL36Aa	ARF	UBC	CACS	TUA	ACT	PP2A	CYP	
BestKeeper 	ARF	TIP41	F-Box	40SRPS8	RPL36Aa	ACT	LEA26	UBC	CYP	CACS	TUA	PP2A	
Normfinder	LEA26	F-Box	40SRPS8	TIP41	RPL36Aa	ARF	UBC	CACS	TUA	ACT	PP2A	CYP	
geNorm	UBC — ARF	F-Box	LEA26	TIP41	RPL36Aa	40SRPS8	CACS	TUA	ACT	PP2A	CYP	
Recommended comprehensive ranking	LEA26	ARF	F-Box	TIP41	40SRPS8	UBC	RPL36Aa	CACS	ACT	TUA	CYP	PP2A	
Ranking Order of candidate reference genes in graft union of different varities of grafted plants (Better–Good–Average)	
Delta CT	TIP41	PP2A	UBC	ARF	40SRPS8	RPL36Aa	LEA26	CACS	ACT	TUA	F-Box	CYP	
BestKeeper 	TIP41	LEA26	PP2A	ARF	UBC	RPL36Aa	ACT	40SRPS8	CACS	F-Box	TUA	CYP	
Normfinder	TIP41	UBC	PP2A	40SRPS8	RPL36Aa	ARF	LEA26	ACT	CACS	TUA	F-Box	CYP	
geNorm	PP2A — ARF	TIP41	40SRPS8	RPL36Aa	UBC	CACS	LEA26	ACT	F-Box	TUA	CYP	
Recommended comprehensive ranking	TIP41	PP2A	ARF	UBC	40SRPS8	LEA26	RPL36Aa	CACS	ACT	F-Box	TUA	CYP	
Ranking Order of candidate reference genes in all samples in grafted cucumber/pumpkin plants (Better–Good–Average)	
Delta CT	CACS	40SRPS8	LEA26	UBC	ARF	TUA	F-Box	ACT	RPL36Aa	TIP41	CYP	PP2A	
BestKeeper 	CACS	LEA26	40SRPS8	TUA	UBC	RPL36Aa	ARF	ACT	F-Box	CYP	TIP41	PP2A	
Normfinder	CACS	40SRPS8	UBC	TUA	LEA26	ARF	F-Box	ACT	RPL36Aa	TIP41	CYP	PP2A	
geNorm	ARF — 40SRPS8	CACS	LEA26	TUA	UBC	ACT	F-Box	TIP41	RPL36Aa	CYP	PP2A	
Recommended comprehensive ranking	CACS	40SRPS8	LEA26	ARF	UBC	TUA	F-Box	ACT	RPL36Aa	TIP41	CYP	PP2A	
Ranking Order of candidate reference genes in all samples (Better–Good–Average)	
Delta CT	CACS	40SRPS8	ARF	UBC	TUA	LEA26	F-Box	ACT	TIP41	RPL36Aa	CYP	PP2A	
BestKeeper	LEA26	CACS	UBC	TUA	40SRPS8	RPL36Aa	CYP	ARF	ACT	TIP41	F-Box	PP2A	
Normfinder	CACS	40SRPS8	ARF	TUA	F-Box	UBC	LEA26	ACT	TIP41	RPL36Aa	CYP	PP2A	
geNorm	CACS — 40SRPS8	ARF	LEA26	UBC	TUA	ACT	F-Box	TIP41	RPL36Aa	CYP	PP2A	
Recommended comprehensive ranking	CACS	40SRPS8	LEA26	ARF	UBC	TUA	F-Box	ACT	RPL36Aa	TIP41	CYP	PP2A	

Expression stability analysis of candidate reference genes

The stabilities of the 12 candidate reference genes in our study were evaluated separately using the ΔCt method, geNorm, NormFinder, BestKeeper, and RefFinder. The 12 candidate reference genes were divided into eight groups of different treatments: organs of cucumber, pumpkin, and cucumber–pumpkin grafted plants under normal conditions were termed Cos, Pos, and Gos, respectively. Graft union samples under low temperatures were termed GLgs, graft union samples during the healing process were termed Ggs, graft union samples of different varieties of cucumber–pumpkin grafted plants were termed Ggvs, all cucumber–pumpkin grafted plant samples were termed GoAll, and all samples in our study were termed All.

ΔCt method analysis

The ΔCt method ranks the stability of expression of tested genes by comparing the relative expression of gene pairs within each sample (Silver et al., 2006). As shown in Table 2, TIP41 was the most stable reference gene in the Cos, Pos, and Ggvs samples, while TIP41 was the lowest stable reference gene in the GLgs samples. CACS was the most stable reference gene in the Gos, GoAll, and All samples, and TUA and L EA26 were ranked as the most stable reference genes in the GLgs and Ggs samples, respectively (Table 2, Table S4).

BestKeeper analysis

The BestKeeper program identifies potential reference genes by calculating the coefficients of variation (CVs) and the standard deviation (SD) of the Ct values, where lower CVs and SD indicate higher stability (Pfaffl et al., 2004). For the Cos and GoAll samples, CACS was identified as the most stable gene, and TIP41 was the most stable gene in the Pos and Ggvs samples. CYP was the most stable gene in the Gos samples, but was the lowest ranking gene in the Ggvs samples. Similarly, TUA was the most stable gene in the GLgs samples, but was the lowest stable gene in the Pos samples. ARF and LEA26 were ranked as the most stable reference gene in the Ggs and All samples, respectively. PP2A was the reference gene with the lowest stability in most samples, including the Gos, Ggs, GoAll, and All samples (Table 2, Table S4).

NormFinder analysis

NormFinder ranks the stability of tested genes based on inter- and intragroup variations in expression across different sample groups, and lower values indicate higher stability (Andersen, Jensen & Orntoft, 2004). TIP4 1 had stability values of 0.084, 0.153, and 0.203, making it the most stable gene in the Cos, Pos, and Ggvs samples, respectively. 40SRPS8 and CACS were the two most stable genes and PP2A was the gene with the lowest stability in the Gos, GoAll, and All samples. For the GLgs samples, TUA was the most stable, but was ranked as the lowest stability reference gene in the Pos samples. The stability of LEA26 was best in the Ggs samples (Table 2, Table S4).

geNorm analysis

The geNorm software determines gene expression stability using M-values based on the average pairwise variation of all candidate genes (Vandesompele et al., 2002). TIP41 and 40SRPS8, CACS and ARF, CYP and UBC, UBC and ARF, PP2A and ARF, ARF and 40SPRS8, were the two most stable genes in the Cos, Gos, GLgs, Ggs, Ggvs, and GoAll samples, respectively. CACS and 40SRPS8 were identified as the most stable reference genes with M-values of 0.093 and 0.582 in the Pos and All samples, respectively (Table 2, Table S4). In addition, the optimal number of reference genes was determined using the geNorm algorithm based on pairwise variation (Vn/Vn+1). A value of Vn/Vn+1<0.15 indicates that the optimal number of reference genes equal to a value of n is sufficient for its use as a reference gene (Vandesompele et al., 2002). In our study, the values of V2/V3 of all experimental samples were less than 0.15, which indicated that two reference genes would be sufficient for gene normalization under these experimental conditions (Fig. 3).

Figure 3 Determination of the optimal number of reference genes.

Pairwise variation Vn/Vn+1 values caculated by geNorm software. A cut-off of 0.15 (Vn value) is usually applied. V1 to V12 stand for the variation in candidate reference genes ranked based on their stability, which V1 is the variation for the most stable and V12 is the least stable gene. Cos: organs of cucumber; Pos: organs of pumpkin; Gos: organs of cucumber-pumpkin; GLgs: graft union of cucumber-pumpkin under low temperature stress; Ggs: graft union of cucumber-pumpkin in healing process; Ggvs: graft union of different varieties of cucumber-pumpkin; GosAll: all grafted cucumber samples; All: all samples.

RefFinder analysis

RefFinder considers the results from the ΔCt method, geNorm, NormFinder and BestKeeper to provide a comprehensive ranking of the most stable genes (Xie et al., 2012). TIP41 was the most stable reference gene in the Cos, Pos, and Ggvs samples, CACS was ranked as the most stable gene in the Gos, GoAll, and All samples. TUA was the most stable in the GLgs samples, but was also the lowest reference gene in the Pos samples. LEA 26 and ARF were the two most stable reference genes in the Ggs samples. PP2A was the lowest stable reference gene in the Gos, Ggs, GoAll, and All samples (Table 2, Table S4).

Validation of the selected reference genes

To confirm the stability of the selected reference genes, the expression levels of four graft union healing-related genes were examined by normalization to the levels of LEA26, ARF, LEA26+ARF, and PP2A as reference genes (Table S3). RefFinder analysis showed that LEA26 and ARF were the most suitable reference genes and PP2A was the least suitable reference gene in the graft union during the healing process (Table 2, Table S4).

The expression patterns of csaCYCD3;1, csaRUL, cmoRUL, and cmoPIN showed similar changes when LEA26, ARF, or LEA26 +ARF were selected as the reference genes for normalization (Fig. 4). The expression levels of csaCYCD and csaRUL were significantly downregulated at 3 d and 6 d compared to 1 d after grafting. However, these values were markedly higher when PP2A was selected for normalization. Compared with the sample 0 day after grafting, cmoPIN expression was clearly downregulated at the graft junction at 6 d, 9 d, and 15 d after grafting when using LEA26, ARF, or LEA26+ARF as reference genes, butthe expression pattern was completely different when PP2A was used as a reference. Similarly, compared with the sample 0 day after grafting, cmoRUL expression levels increased 3.99, 3.31, and 3.43 times at the graft junction 3 days after grafting when using LEA26, ARF, and LEA26+ARF as reference genes, respectively, but the change in expression was calculated as a 12.70-fold increase if the level of PP2A was used for transcript normalization.

Figure 4 Relative expression levels of csaCYCD3;1 (A), cmoPIN (B), csaRUL (C), cmoRUL (D) using different reference genes at the graft union at 0, 1, 3, 6, 9, 15d after grafting.

The two most suitable reference genes (LEA, ARF), their combination (LEA26 +ARF), and the least stable reference gene (PP2A) by RefFinder analysis were used for expression normalization. Bars represent the means and standard deviations of three biological replicates.

Discussion

qRT-PCR is a powerful method for detecting transcriptomic data and studying underlying molecular mechanisms (Niu et al., 2017). Appropriate reference genes are required to ensure the accuracy of qRT-PCR results. There has been research into the selection of optimal reference genes in cucumber and pumpkin (Wan et al., 2010; Obrero et al., 2011; Warzybok & Migocka, 2013), however, there have been no studies on the selection of the optimal reference genes for cucumber–pumpkin grafted plants. Grafting assembles the scion and rootstock into a plant with advantages over the parent plants. This is significant exchange of materials between scion and rootstock, includingwater, sugars, hormones, RNAs, and proteins (Melnyk, 2017), so it is highly likely that the optimal reference genes for cucumber–pumpkin grafted plants may differ from the optimal reference genes for cucumber or pumpkin. To test this, we selected some commonly used reference genes that are previously published (ACT, CYP, CACS, TUA, TIP41, F-Box, PRL36Aa and PP2A) and are known to be expressed in cucumber or pumpkin. We also selected four novel genes (UBC, ARF, LEA26 and 40SRPS8) from our transcriptomic data on graft union healing in cucumber–pumpkin grafted plants, and primers were designed based on the conserved sequence of the genes between cucumber and pumpkin.

The ΔCt method, BestKeeper, NormFinder, geNorm, and RefFinder are five software programs and methods that are commonly used to identify reference genes (Scarabel et al., 2017; Duan et al., 2017). In our study, two genes were sufficient for reliable normalization when all samples were subjected to geNorm analysis (Fig. 3). The ΔCt method, NormFinder, geNorm, and RefFinder programs all suggested the same least suitable reference genes, but differed from the rankings obtained by BestKeeper. For instance, F-Box was ranked as the least stable gene in Cos samples by the ΔCt method, NormFinder, geNorm, and RefFinder programs analysis, while BestKeeper identified UBC as the lowest stability gene in the Cos samples. This different result for different methods is consistent with the result of a study by Niu et al. (2017) where the rankings obtained by BestKeeper were also different from those obtained by geNorm and NormFinder. The most suitable reference genes differed between the five algorithms. Six of the traditional reference genes (TIP41, CACS, ARF, UBC, CYP and PP2A) and two novel reference genes (LEA26 and 40SRPS8) were identified as the optimal reference genes in different samples using different algorithms for analysis. The comprehensive evaluation by RefFinder used data from the other four computational methods, and this ranking showed that TIP41, CACS, TUA, and LEA26 were the most suitable reference genes in different samples of cucumber, pumpkin, and cucumber–pumpkin grafted plants.

TIP41 is a tonoplast intrinsic protein that functions as a PPA2 activator in plants, and has been identified as a suitable reference gene in Cucumis sativus (Wan et al., 2010), Cichorium intybus (Delporte et al., 2015), and Papaver rhoeas (Scarabel et al., 2017). In our study, TIP41 was a stable reference gene in cucumber, pumpkin, and at the graft union of different varieties of grafted cucumber plants, but TIP41 was ranked as a gene with relatively lower stable in the Gos samples (Table 2). This may suggest that grafted plants are different from the nongraft scion and rootstock at the molecular level. Surprisingly, TIP41 was ranked as the least stable reference gene in the graft union of cucumber–pumpkin grafted plants at low temperatures. Importantly, reference gene stability can vary under different experimental treatments (Bustin et al., 2005). Reid et al. (2006) showed that TIP41 is an inadequate reference gene during berry development. Similarly, TUA was regarded as the most stable reference gene in the graft union under cold stress, but was ranked as the least suitable reference gene in pumpkin organs by RefFinder analysis (Table 2). In cucumber, TUA is a highly stable gene when different tissues were treated with abscisic acid, salicylic acid, and methyl jasmonic acid (Wan et al., 2010), however, TUA was limited as a stable reference gene in cucumber under conditions of salt, osmotic stress, and high or low temperature (Wan et al., 2010; Migocka & Papierniak, 2011). CACS encodes the clathrin adaptor complex subunit, which links clathrin to receptors in vesicles (Migocka & Papierniak, 2011). As this gene participates in a basic intracellular transport process, CACS has been used as a reference gene at different developmental stages and under varying environmental conditions in Arabidopsis thaliana (Czechowski et al., 2005), buckwheat (Fagopyrum esculentum) (Demidenko, Logacheva & Penin, 2011), and lettuce (Lactuca sativa) (Borowski et al., 2014). In cucumber, CACS was ranked as the best reference gene under different nitrogen nutrition conditions (Warzybok & Migocka, 2013), heavy metal stress, and on deprivation and/or readdition of different nutrients (N, C, P, and S) (Migocka & Papierniak, 2011). One of the novel reference genes, LEA26 (Late Embryogenesis Abundant protein 26), is related to abiotic stress tolerance, especially desiccation tolerance in Arabidopsis. LEA26 has not been evaluated as a reference gene in any species (Dang et al., 2014). In our study, LEA26 was the most stable reference gene in the Ggs. However, LEA26 was also identified as the lowest stable in the GLgs samples by BestKeeper analysis and exhibited relatively lower stability in the Pos sample. Overall, the results showed it was necessary to validate reliable reference genes prior to qRT-PCR analysis under detailed experimental conditions.

To validate use of the identified reference genes as control genes with unchanging expression levels, the expression levels of csaRUL, csaCYCD3;1, cmoRUL, and cmoPIN in the cucumber-pumpkin graft union healing process were normalized by the two most stable reference genes and the least stable gene. The results showed that LEA26 and ARF may be the best candidate reference genes for the normalization of gene expression in the graft union healing process. The use of inappropriate reference genes may lead to inaccurate results, making it extremely important to identity suitable reference genes to increase the reliability of qRT-PCR data for target gene expression.

Conclusions

To our knowledge, this is the first report of the simultaneous use of cucumber, pumpkin and their grafted plants as samples to identify optimal candidate reference genes. Twelve candidate reference genes were validated in different organs, conditions, species of cucumber, pumpkin and their grafted plants using five software tools-the ΔCt method, BestKeeper, NormFinder, geNorm and RefFinder. The results showed that TIP41 and CACS showed the most stable expression in different cucumber organs, TIP41 and PP2A were the optimal reference genes in pumpkin organs, TUA and RPL36Aa were the most stable in the graft union under cold stress, LEA26 and ARF showed the most stable expression in the graft union during the healing process, TIP41 and PP2A were the most stable across different varieties of cucumber–pumpkin grafted plants, and CACS and 40SRPS8 were the most stable in all grafted cucumber samples. Our analysis showed that two genes are sufficient for reliable normalization when all samples are considered. This work should facilitate future studies on gene function and molecular mechanisms in cucumber–pumpkin grafted plants and other closely related species.

Supplemental Information

Table S1 Summary for the graft union transcriptome

This is description of transcriptiome data of graft union at 0, 3, 6, 9 days after grafting.

Click here for additional data file.

Table S2 RPKM values of the 10 cucumber genes and 7 pumpking genes covering transcriptomes data of graft union at the 0d (ck-m), 3d, 6d, 9d after grafting

This is raw data from the transcriptomes data.

Click here for additional data file.

Table S3 RPKM values of the csaCYCD3;1, csaRUL, cmoRUL, cmoPIN genes and the commonly used refenence genes covering transcriptomes data of graft union at the 0d (ck-m), 3d, 6d, 9d after grafting

This is raw data from the transcriptomes data.

Click here for additional data file.

Table S4 Genes expression stability ranked by RefFinder, Delta CT, Bestkeeper, Normfinder, geNorm

Click here for additional data file.

Table S5 GO terms of reference genes

Click here for additional data file.

Figure S1 Illustrations of hole insertion grafting methods process in cucumber grafted on pumpkin

A cucumber cultivar (Zhongnong No.26) was used as the scion, a pumpkin cultivar (Jinxinzhen No.5) was used as the rootstock. The rootstocks were sown 2–3 d earlier than scions (6–7 d after sowing ). When cotyledons of the scion were fully opened and the first true leaf of the rootstock started to develop (9–10 d after sowing) plants grafted as previously described (Mohamed et al., 2014). A hole on the upper portion of the rootstock hypocotyls was made, and then the growing point of the rootstock were removed with a razor blade. The scion was cut on a 30 ° -60 ° on both sides of the hypocotyls, then made the scion insert into the hole made in the rootstock quickly, and the cut surfaces were matched together and held with a grafting clip (Fig. S1).

Click here for additional data file.

Figure S2 Amplification of single PCR product of the expected size for 12 candidate reference genes using cucumber (A), pumpkin (B), cucumber-pumpkin grafted plants. Lines: 1-10, ACT, CYP, CACS, TUA, TIP41, F-Box, PRL36Aa. A

Based on the conserved sequence of these genes between cucumber and pumpkin, primers were designed using Primer Premier 5.0 software with the following parameters: a melting temperature (Tm) of 50–60° C, a primer length of 17–25 bp, and a product size of 70–260 bp (http://www.premierbiosoft.com/) (Table 1). Amplification of a single PCR product in 1% agarose gel electrophoresis.

Click here for additional data file.

Figure S3 Melting curves of 12 candidate reference gene in cucumber (A) and pumpkin (B)

The Fluorescence changes (X-axis) was plotted versus the reaction temperature of qRT-PCR (Y-axis). A single peak of the melting curve in qRT-PCR were used to ensure the specificity of the primers for the candidate reference genes.

Click here for additional data file.

Figure S4 Melting curves (A) and Standard curves (B) of 12 candidate reference genes in cucumber-pumpkin grafted plants

The Fluorescence changes (Y-axis) was plotted versus the reaction temperature of qRT-PCR (X-axis). A single peak of the melting curve in qRT-PCR were used to ensure the specificity of the primers for the candidate reference genes (Fig. S4A).The amplification efficiencies for each primer and the regression coefficients (R2) were evaluated using five-fold dilutions of pooled cDNA (1/5, 1/25, 1/125, 1/625, 1/3125) that were diluted using EASY dilution solution, the Ct values changes (Y-axis) was plotted versus the initial quality of qRT-PCR (X-axis). The linear regression equation of every primer was also showed in Fig. S4B.

Click here for additional data file.

Figure S5 Standard curves of 12 candidate reference genes in cucumber (A) and pumpkin (B)

The amplification efficiencies for each primer and the regression coefficients (R2) were evaluated using five-fold dilutions of pooled cDNA (1/5, 1/25, 1/125, 1/625, 1/3125) that were diluted using EASY dilution solution. The Ct values changes (Y-axis) was plotted versus the initial quality of qRT-PCR (X-axis. The linear regression equation of every primer was also showed in Fig. S5.

Click here for additional data file.

Supplemental Information 1 1 Ct values of samples in Cos, Pos, Gos,GLgs, Ggs, GosALL, and All groups. 2 Determination of the optimal number of references genes

1 Ct values of samples in Cos, Pos, Gos,GLgs, Ggs, GosALL, and All groups. All samples were performed with three biological and technical replicates. Cos: organs of cucumber; CL and CR were the abbreviation for leaf, stem, root of cucumber, recpectively.Pos: organs of pumpkin; PL and PR were the abbreviation for leaf, stem, root of cucumber, respectively.Gos: organs of cucumber/pumpkin grafted plants; GL, GGS, GPS, and GR were the abbreviation for leaf, stem of scion, stem of graft union, stem of rootstock, and root of grafted cucumber, respectively.Ggs: graft union of cucumber/pumpkin in healing process; samples were harvested at 0d, 1d, 3d, 6d, 9d, 15d after grafting.Ggvs: graft union samples of different varieties of cucumber–pumpkin grafted plants; XZ, XJ, XH, ZJ, ZH were the abbreviation for‘Xintaimici–Zhongguonangua No. 26’, ‘Xintaimici–Jinxinzhen No. 5’, ‘Xintaimici–Huofenghuang’, ‘Zhongnong No. 26–Jinxinzhen No. 5’, and‘Zhongnong No.26–Huofenghuang’ grafting combinations, respectively.All: all samples

2 Determination of the optimal number of references genes. Pairwise variation Vn/Vn+1 values calculated by geNorm software were used to determine the optimal number of reference genes. A cut-off of 0.15 (Vn value) is usually applied.

Click here for additional data file.

We are particularly appreciated to Mengmeng Duan, Jinglei Wang, and other members in our research group for their kind suggestions to perfect the experiment.

Additional Information and Declarations

Competing Interests

Author Contributions

Data Availability

The authors declare there are no competing interests.

Li Miao performed the experiments, contributed reagents/materials/analysis tools, prepared figures and/or tables, authored or reviewed drafts of the paper.

Xing Qin analyzed the data, prepared figures and/or tables.

Lihong Gao contributed reagents/materials/analysis tools, authored or reviewed drafts of the paper.

Qing Li analyzed the data.

Shuzhen Li performed the experiments.

Chaoxing He contributed reagents/materials/analysis tools.

Yansu Li and Xianchang Yu conceived and designed the experiments, approved the final draft.

The following information was supplied regarding data availability:

The raw measurements are available in File S2.

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
