# Peer review of "Selection of reference genes for quantitative real-time PCR analysis in cucumber (Cucumis sativus L.), pumpkin (Cucurbita moschata Duch.) and cucumber–pumpkin grafted plants"

_PeerJ, doi:10.7717/peerj.6536_

## Round 0.1 · original submission · Major Revisions

Dear Author,

Concerning the detailed reviews of the reviewers, it is essential to make a clear point-by-point revision in order to improve your manuscript.

Best regards

Reviewer 1 ·

Basic reporting

Well done.
Minor:
Lines 58/59: Formatting error
Line 112: Please add the authors for all species you mention in the complete text not only for Cucumis sativus.
Formatting of the of the reference genes (in Italics) has to be carefully checked in the complete manuscript. There are more formatting issues that need to be carefully corrected.
Reference in line 486 is not in alphabetical order. There is a disproportionally high number of references published by Asian/Chinese authors. There are many more relevant publications from authors from all over the world. Please carefully address this point.

Experimental design

Abstract
The abstract provides all information in a condensed way.

Introduction
In the introduction many studies mentioning many putative reference genes are listed. What could be added is a more detailed compilation of the problems that might occur when the reference genes chosen are not experimentally verified. This aspect is still not taken into account in many publications which often leads to the uncritical application of Actin as a reference gene for many different cultivation conditions.

Material & Methods
Line 154/155: Please be more specific about the unpublished data. It might be more scientifically meaningful to publish the unpublished data set together with the analysis of the reference genes submitted in this manuscript.

Results
The paragraph starting in line 222 is very long and need to be structured. In additions some sentences could be written in a clearer way. Please carefully improve this paragraph for better readability.

Discussion
Part of the discussion is a repetition of aspects mentioned already in the other parts. Please try to reduce and be more specific with respect to your data.

Conclusions
This important part of paper needs to be more carefully arranged and completely rewritten because of incomplete sentences and repetitions.

Supplementary
The supplemental data files need meaningful legends, best summarized in a legend file. The note in Table S1 is not clear and the sentence and the described fast hast to be described in more detail.

Validity of the findings

The data are well presented and the description and conclusion based on the data are plausible and evident.

Additional comments

The authors carry out a very broad study investigating 12 different reference genes. There are no technical concerns on the execution of this study. However, this paper is a very technical paper as the authors also state in their cover letter. My major concern is that it could be part as a supplement of a publication that deals already with a real biological research question instead of being published by its own.

Reviewer 2 ·

Basic reporting

Li et al. reports the optimal reference gene to normalize the expression data for qRT-PCR in cucumber, pumpkin and cucumber-pumpkin grafted plants by four statistical tools. Eight candidate genes were tested under various conditions, and most constant expression genes were defined as candidate reference genes. This study is a well-organized and thorough to identify optimal reference genes. However, the current manuscript is missing several important requirements for data presentation and methodology explanation. I provide major and minor points to revise.

Experimental design

No problem.

Validity of the findings

There is no large impact but meaningful.

Additional comments

Li et al. reports the optimal reference gene to normalize the expression data for qRT-PCR in cucumber, pumpkin and cucumber-pumpkin grafted plants by four statistical tools. Eight candidate genes were tested under various conditions, and most constant expression genes were defined as candidate reference genes. This study is a well-organized and thorough to identify optimal reference genes. However, the current manuscript is missing several important requirements for data presentation and methodology explanation. I provide major and minor points to revise.

<Major comments>
1. In my view, I do not find any citation of Figs 3 and 4 in the text. Please remove or cite them in the text.
2. Figure titles and legends are not written in proper manner. Some parts seems to be just copied, then titles include detailed panel information and legends are missing information needed to understand all presented figure panels. Please consider to revise them.
3. As the current study screened available references which have stable expression patterns under several stressful conditions, the authors could consider to inform such strategy in titles and introduction section.

<Point-to-point comments>
L167; don’t→do not
L177; H20→H2O
L186-187; This part should be explained in result section.
L208; It is the content of the method. Please move this sentence to the method section.
L224−233 and L310-312; These seem to be the content of the introduction. Please remove or explain them in introduction section.
L346; Here, some reference is required.
Gene names/transcripts should be written in italics, eg L77-78, L81-82, and L355. Please check throughout the manuscript and revise them.

·

Basic reporting

Miao et al. report on the evaluation of a reference gene set for the use in cucumber, Pumkin and cucumber-pumpkin grafted plants. The experimental procedure is sound and well described in good English language (judged as non-native speaker). Results, discussion, tables, figures and supplemental material are well written and documented. The raw data are shared. Literature is mostly documented (for missing references see below). The whole paper is well structured.
Authors refer to their own unpublished transcriptomic data (L155). This should be publishes before or along with the manuscript.

Experimental design

The experiments are sound and well described with some minor comment detailed below.

Validity of the findings

The findings are valid.

Additional comments

I have some comments on different passages of the text:

Title: delete "optimal". There is no proof that these are the best. Only for the current conditions, they may be the most suited reference genes.

Title (L3): include ...gene expression "data derived from" instead of gene expression in cucumber

L45: delete "optimal"

Introduction: Sentence and reference missing on the MIQE guidelines: Bustin SA et al. (2009) The MIQE Guidelines: Minimum Information for Publication of Quantitative Real-Time PCR Experiments Clin Chem 55:611-622 doi:10.1373/clinchem.2008.112797

L52: replace "always" by "usually"

L81-86: Unclear. The mentioned reference genes were used all together or one by one? Explain the "specific environments"

84: italics "Cucurbita pepo"

L92-97: name and function of the mentioned genes

L103: replace "gernerated" by "generated"

L104-107: provide reference after "...which are thought to be related to graft union healing in grafted cucumber..."

L117: give size or volume of the pot instead of cell number

L122: ...previously described (Fig S1). Please provide reference.

L139: replace "seedlings" by "grafted plants"

L152: specify "other plants"

Why did authors not evaluate the "classic" reference genes in their transcriptomic data? This is missing.

L161: BLAST against what data base?

L163: rephrase sentence to "This resulted in ten and seven genes of cucumber and pumkin, respectively, which may be suitable as reference genes"

L168: provide reference for Primer Premier 5.0 software

L174: replace "the" by "an"

L176: give manufacturer of "Premix DimerEraser"

L193: Please provide reference instead of web page

L205: "eight genes used traditionally...and four potential (instead of "new") reference genes..."

L232: From here onward many abbreviations are used. These should be limited and better explained.

L241: Explanation of V2/3 value missing. Please add.

L222: The whole section should be shortened and summarized with a table if possible.

L299 & 300: "abnormally" is not a scientific expression. Please replace by proper term.

L338: replace "," with "."

L364: "Grafted plant is..." has nothing to do with the context of the manuscript. Delete.

L362: Conclusion: Authors did not comment on the importance of validation of reference genes for each experimental setting, organ, or treatment. Please add.

---

## Round 0.2 · Minor Revisions

Dear Author,

As all reviewers stated minor revisions, it is clear that you improved the manuscript thoroughly content-wise. Nevertheless, a major concern related to the English language remains and needs to be addressed in the following revision in order to become acceptable for publication.
Success,

Ann

Reviewer 1 ·

Basic reporting

The authors tried hard to improve according to the suggestions of the reviewers. In some aspects it might be too verbose now. Please try to focus on the most important aspects and shorten te results section.
The English language has to be corrected by a native speaker.
In my version are many blanks missing, mainly in the part added in red. It might be a version problem but has to be solved.

Experimental design

There are no technical concerns on the execution of this study.

Validity of the findings

There are no technical concerns on the execution and evaluation of this study.

Additional comments

My major concern is still that it could be part as a supplement of a publication that deals already with a real biological research question instead of being published by its own. The arguments of the authors are not convincinb to me.

Reviewer 2 ·

Basic reporting

Thank you for the resubmission. I feel written texts are well proofreading, and very close to accept for publication. However, there are still some points that require revision, so please correct them.

Experimental design

No problem.

Validity of the findings

There is no large impact but meaningful.

Additional comments

Over the entire paper, there were several points that did not have the required space, such as before and after the parentheses or after the comma. Please insert correct spaces.

Figure titles of Fig. 1, 2, and 3 contain the contents of figure legend. Please remove the contents of the figure legend.

L202; Fig2 is the content of the result, that it is not suitable to be in Materials & Methods.

·

Basic reporting

This is the review of the revised version of the manuscript. Most comments are presented in the attached documents. Here, only some general comments:
1. many spaces between words missing.
2. Manuscript needs careful English editing.

The manuscript is not ready for publication in its current state. I suggest a minor revision.

Experimental design

na

Validity of the findings

na

Additional comments

na

---

## Round 0.3 · accepted · Accept

Congratulations with the improvement of the revision of the manuscript.

#